# POLAR PROBE LINEARLY DECODES SEMANTIC STRUCTURES FROM LLMS

## ABSTRACT

How do artificial neural networks bind concepts to form complex semantic structures? Here, we propose a simple neural code, whereby the existence and the type of relations between entities are represented by the distance and the direction between their embeddings, respectively. We test this hypothesis in a variety of Large Language Models (LLMs), each input with natural-language descriptions of minimalist tasks from five different domains: arithmetic, visual scenes, family trees, metro maps and social interactions. Results show that the true semantic structures can be linearly recovered with a Polar Probe targeting a subspace of LLMs' layer activations. Second, this code emerges mostly in middle layers and improves with LLM performance. Third, these Polar Probes successfully generalize to new entities and relation types, but degrades with the size of the semantic structure. Finally, the quality of the polar representation correlates with the LLM's ability to answer questions about the semantic structure. Together, these findings suggest that LLMs learn to build complex semantic structures by binding representations with a simple geometrical principle.

## 1 INTRODUCTION

**Compositional representations.** Human languages constantly require combining words into rich semantic structures – such as family ties, spatial arrangements, or part-whole connections. Consider the sentence '*Bob is Alice's father, and Mary is her mother*': understanding it requires constructing, on the fly, a representation of the underlying family tree. Yet, simply allocating a one-hot feature for each possible combination quickly becomes impractical, and indeed prevents generalization (Fodor & Pylyshyn, 1988). As large language models become increasingly able to combine new concepts (Brown et al., 2020), it is thus critical to understand *how* the geometry of their activations bind entities to represent compositional structures.

**Probing syntactic structures.** A simple binding principle has recently been evidenced in the context of syntactic representations. Indeed, Hewitt & Manning (2019) showed with a *Structural Probe* that words that are linked syntactically (e.g., subject - verb) are represented more closely in a specific subspace of the LLMs' activations than words that aren't (e.g., subject - object). Building on this proposal, Diego-Simon et al. (2024) further showed with a *Polar Probe* that the *type* of syntactic relation can be recovered from the relative *direction* between the two words in this subspace. In sum, the structure of syntax can be explicitly represented through the relative distances and directions between contextualized word embeddings.

**Remaining challenge.** This binding principle, however, is currently limited to *syntax* (Müller-Eberstein et al., 2022; Eisape et al., 2022; Limisiewicz & Mareček, 2021). Consequently, it is unclear whether a similar principle may also be at play in *semantic* binding.

**Approach.** To test this hypothesis, we evaluate whether LLMs build subspaces of activations where the relative distance and direction between word embeddings linearly represent the corresponding semantic structure. To evaluate the generality of our approach, we introduce a synthetic dataset spanning five semantic domains: variable ordinality, spatial layouts, thematic roles, family trees and metro maps, each with distinctive properties. Each sample in the dataset consists of a text that describes a semantic structure (e.g., *The dog is to the left of the cat. The cat is below the table.* etc).

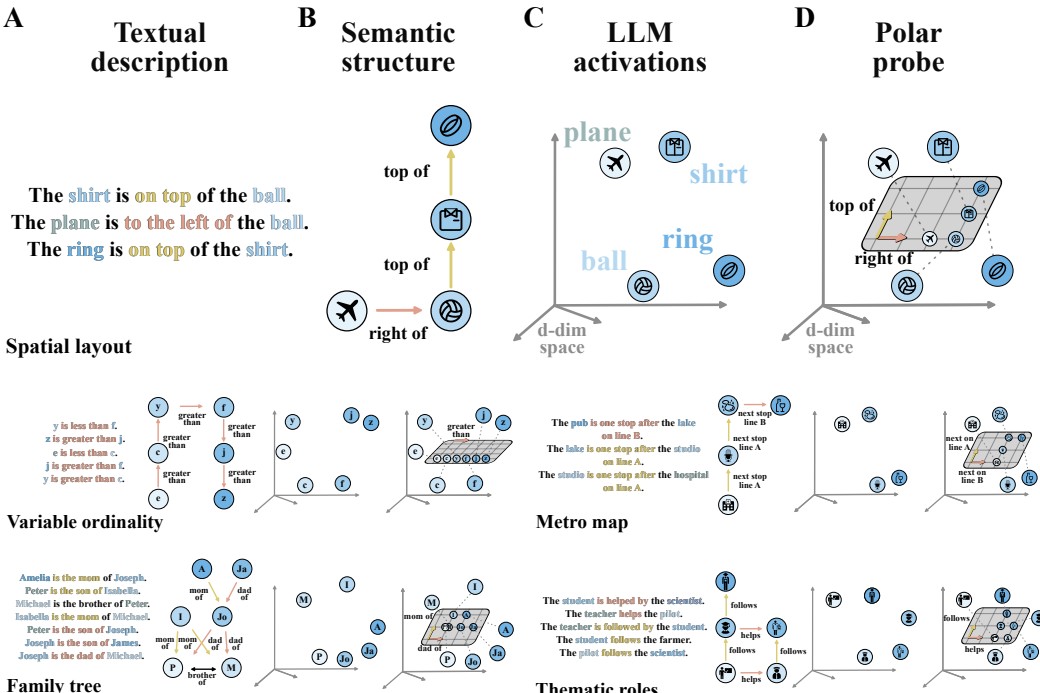

Figure 1: **Polar probes linearly read out semantic structures from LLM activations.** **A:** A natural-language description specifies a set of entities and their typed relations (illustrated here for *spatial layout*, where entities are objects and relations are spatial predicates (left of/right, top of/ below). **B:** The description corresponds to a semantic structure, formalized as a relational graph whose nodes are entities and whose edges are typed, directed relations. **C:** The LLM contextualizes each entity token, yielding a high-dimensional entity representation in its residual stream. **D:** A polar probe, a learned linear transformation from activation space to a probe-space recovers the relational graph: Euclidean distance between entity representations codes for edge *existence*, while relative direction codes for relation *type*. **Bottom panels:** The same scheme applies to four additional domains: spatial layouts, family trees, metro maps, and thematic roles.

For each semantic structure, we generate multiple valid *textual descriptions*, where we randomize the order of relations and entities, to break any correlation between word order and graph structure. We then input these text descriptions to a variety of LLMs, differing in size and pretraining stage, and train a Polar Probe (Diego-Simon et al., 2024) on their activations, one per layer and per domain. To verify that the identified representations effectively generalize, we evaluate out-of-domain (OOD) samples with new entity names and relation surface forms. Finally, to evaluate the functional role of the polar probe, we test whether the quality of these polar representations predicts the LLMs' ability to answer questions about the semantic structure.

## 2 METHODS

### 2.1 PROBLEM FORMALIZATION

**Semantic structures as directed graphs.** Semantic structures can be formalized as *relational graphs* with (i) labeled nodes and (ii) directed and labeled edges, where entities are the nodes, and semantic relations are the edges.

Let $\mathcal{V} = \{v_1, v_2...v_{|\mathcal{V}|}\}$ be a finite set of entities and $\mathcal{T} = \{t_1, t_2...t_{|\mathcal{T}|}\}$ a finite set of relation types. Define the set of all typed, directed edges without self-loops:

$$\mathcal{E} = \{(v_i, v_j, t_r) \in \mathcal{V} \times \mathcal{V} \times \mathcal{T} : i \neq j\}.$$

A semantic structure can thus be formalized as $G = (\mathcal{V}_G, \mathcal{E}_G, \mathcal{T}_G)$, where $\mathcal{V}_G \subseteq \mathcal{V}$, $\mathcal{T}_G \subseteq \mathcal{T}$ and $\mathcal{E}_G := \{(v_i, v_j, t_r) \in \mathcal{E} \mid v_i, v_j \in \mathcal{V}_G, t_r \in \mathcal{T}_G\}$.

Such graph can be written in an algebraic form: First, the undirected pairwise distance matrix $M_G^\rho \in \mathbb{N}_0^{|\mathcal{V}_G| \times |\mathcal{V}_G|}$ is:

$$(M_G^\rho)_{ij} = d_G(v_i, v_j)$$

where $d_G : \mathcal{V}_G \times \mathcal{V}_G \to \mathbb{N}_0$ denotes the (shortest-path) graph distance. Second, the target incidence tensor $M_G^\phi \in \{-1, 0, 1\}^{|\mathcal{V}_G| \times |\mathcal{V}_G| \times |\mathcal{T}|}$ one-hot-encodes relation types:

$$(M_G^\phi)_{ijr} = \begin{cases} 1, & (v_i, v_j, t_r) \in \mathcal{E}_G, \\ -1, & (v_j, v_i, t_r) \in \mathcal{E}_G, \\ 0, & \text{otherwise.} \end{cases}$$

**Semantic structures as LLM activations.** A semantic structure $G$ can also be described in natural language. For simplicity, we consider that each entity $v_i \in \mathcal{V}_G$ is represented by a unique token $w_i$. If an LLMs input with such a textual description represents $G$, then this structure should be retrievable from its hidden activations $\mathbf{h}_i \in \mathbb{R}^d$ (Vaswani et al., 2017).

In sum, we seek to identify how the symbolic/graphical representations of semantic structures are represented in the vectorial activations of neural networks.

**Polar Probe.** Our hypothesis is that LLMs represent $G$ through a Polar Coordinate system: i.e. a subspace of $h$, where related nodes are relatively close to one another, and their relative direction is specific to their relation type (Diego-Simon et al., 2024).

In practice, the Polar Probe is trained in a supervised manner to find the linear transformation $B \in \mathbb{R}^{k \times d}$ (with $k \leq d$) of the entity embeddings to best approximate the true pairwise distance matrix $M_G^\rho$ and the incidence tensor $M_G^\phi$. For a pair of entities $(w_i, w_j)$, we define their probed relation as $\boldsymbol{\delta}_{ij} = \mathbf{z}_i - \mathbf{z}_j$, with $\mathbf{z}_i = B\mathbf{h}_i \in \mathbb{R}^k$.

The probed pairwise distance matrix $\hat{M}_G^\rho \in \mathbb{R}^{|\mathcal{V}_G| \times |\mathcal{V}_G|}$ is computed as:

$$(\hat{M}_G^\rho)_{ij} = \|\boldsymbol{\delta}_{ij}\|_2,$$

where $\|\cdot\|_2$ is the $\ell_2$-norm.

To assign each relation to a specific direction of the probed space, we learn a prototype vector $\mathbf{p}_r$ for each relation type $r \in \mathcal{T}$ and train the probe so that, whenever a relation of type $r$ holds between $(v_i, v_j)$, the probed difference $\boldsymbol{\delta}_{ij}$ aligns with $\mathbf{p}_r$. The probed incidence tensor $\hat{M}_G^\phi \in \mathbb{R}^{|\mathcal{V}_G| \times |\mathcal{V}_G| \times |\mathcal{T}|}$ records, for each token pair $(w_i, w_j)$, the cosine similarity between $\boldsymbol{\delta}_{ij}$ and every prototype vector:

$$(\hat{M}_G^\phi)_{ijr} = \frac{\boldsymbol{\delta}_{ij} \cdot \mathbf{p}_r}{\|\boldsymbol{\delta}_{ij}\|_2 \|\mathbf{p}_r\|_2}.$$

**Learning objective.** Consequently, the loss of the Polar Probe consists of two loss terms.

The structural loss $\mathcal{L}_s$ (Hewitt & Manning, 2019) is:

$$\mathcal{L}_s = \frac{1}{|\mathcal{B}|} \sum_{G \in \mathcal{B}} \left(1 - \Psi(\hat{M}_G^\rho, M_G^\rho)\right)$$

$\Psi(\cdot, \cdot)$ is the differentiable Spearman rank correlation over the vectorized upper-triangular entries of the two distance ($M_G^\rho$ and $\hat{M}_G^\rho$); ranks are computed via the Sinkhorn-based soft-sorting operator of Blondel et al. (2020). $\Psi$ assesses agreement in relative distance orderings (higher is better) and does not assume a common scale or equal-interval spacing between distances.

The angular loss $\mathcal{L}_a$ (Diego-Simon et al., 2024) is

$$\mathcal{L}_a = \frac{1}{|\mathcal{B}|} \sum_{G \in \mathcal{B}} \frac{1}{|E_G| |\mathcal{T}|} \sum_{(i,j) \in E_G} \sum_{r \in \mathcal{T}} \left((\hat{M}_G^\phi)_{ijr} - (M_G^\phi)_{ijr}\right)^2$$

The polar probe and the relation prototypes are jointly trained to minimize a weighted ($\lambda in \mathbb{R}$) sum of the structural ($\mathcal{L}_s$) and angular ($\mathcal{L}_a$) losses:

$$(B^*, \{\mathbf{p}_r^*\}_{r \in \mathcal{T}}) = \underset{B, \{\mathbf{p}_r\}_{r \in \mathcal{T}}}{\arg\min} \left( \mathcal{L}_s + \lambda \mathcal{L}_a \right), \quad \lambda > 0.$$

**Implementation details.** For simplicity, we systematically train and evaluate the Polar Probe independently on each of the layers of a given LLM. [1] After a grid search, we set $\lambda = 5.0$ and the learning rate to $1 \times 10^{-5}$. Training runs for 100 epochs, and the probe rank is 512 by default unless stated otherwise. For each semantic domain, the training set contains 30 graphs, each described in 20 distinct ways. The validation set contains 50 unseen graphs, each with described 20 times. All graphs in a dataset have a fixed number of entities; entity names are drawn from a pool of 13 items. A disjoint pool of 13 entities and relation surface forms is reserved for OOD evaluation.

## 2.2 DATASETS

**Semantic domains.** We synthesize five datasets, each containing relational graphs from a different semantic domain. We group semantic domains into *Euclidean* (spatial layouts, variable ordinality and thematic roles) and *non-Euclidean* (metro networks and family trees). The criterion is whether the graphs can be faithfully coded with a polar code in a flat (zero–curvature) space, or instead require nonzero curvature.

**Prompting.** We use a short, domain-specific prompt to introduce the LLM to the semantic domain and to incline it to infer the relations between entities. For more details refer to table 1. For example, (e.g., *I am going to describe a family tree, you need to understand the how all family members relate to each other*). After this prompt, the relations in the graph are exhaustively described in a random order. For each relation, the order of the entities is also randomized between both equivalent options (e.g., *James is the dad of Joseph* or *Joseph is the son of James*). Therefore, for a given relational graph, there exists multiple textual descriptions.

**Post-prompt.** Entities that are related tend to be relatively close in the textual descriptions, because we do not have any sentences that describes lack of relations. Consequently, to ensure that the distance between probed entities is not confounded by the distance between words, we append the list of entities in a fixed order (e.g., *Who are James, Joseph, and Amelia?*) and use those as probed tokens. To guarantee a one-to-one mapping between entities and tokens, entity names where chose to ensure that they are coded by a single-token.

### 2.2.1 EUCLIDEAN GRAPHS.

We generate Euclidean relational graphs for three semantic domains via Algorithm 1.

**Arithmetic (Variable ordinality).** First, we investigate ordinality of mathematical variables, such semantic domain consists of a single relation type. Different variable names are randomly placed on a magnitude axis. Relational graphs are constructed where entities are mathematical variables (e.g., $x$, $y$, $z$) and relations denote relative order between adjacent variables (e.g., greater than/less than).

$$\text{``} x \text{ is greater than } z. \ z \text{ is less than } y.\text{''}$$

**Spatial arrangement (2D layouts).** Second, we investigate spatial layouts in two dimensions. Such layouts consist of placing objects on a two-dimensional regular grid. Relational graphs are constructed where entities are objects (e.g., *ball*, *shirt*, *plane*) and relations denote relative position between adjacent objects (e.g., left of/right of; above/below).

$$\text{``The } ball \text{ is left of the } plane. \text{ The } ball \text{ is below the } shirt.\text{''}$$

**Social interaction (Thematic roles).** Third, we turn to abstract social interactions, considering only cases where the relational graph admits an exact two-dimensional Euclidean embedding under a polar code. We model thematic roles that capture agent–patient interactions within a group of

---

[1]Code and data will be made publicly available with the camera-ready.

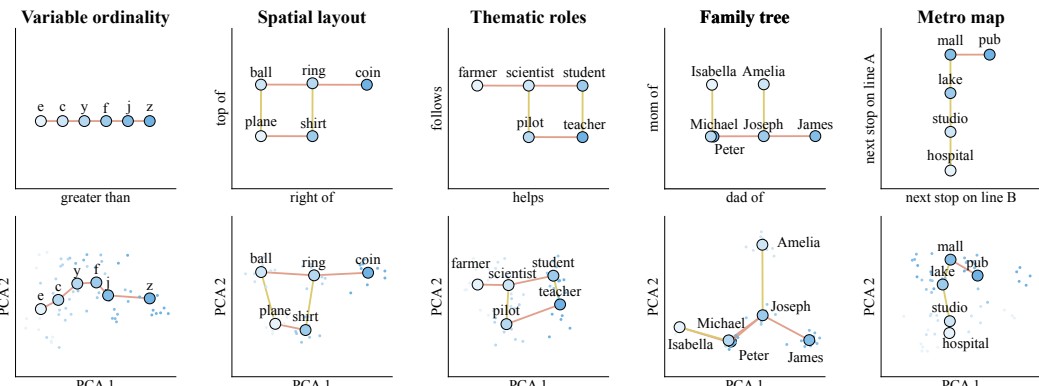

Figure 2: **Polar probe geometry mirrors the gold semantic structure. Top:** Expected polar probe geometry for semantic structures from every domain. **Bottom:** 2D PCA of probe-space entity representations from 10 different descriptions of a semantic structure in the test set; large markers denote entity centroids and lines indicate gold relations. The projections tend to follow the polar code: direction encodes relation *type*, and Euclidean distance encodes relation *existence* and proximity between entities in the relational graph.

people. Relational graphs are constructed where entities are professionals (e.g., *farmer*, *pilot*, *teacher*), and relations denote interaction types between an agent and a patient (e.g., follows/followed by; helps/helped by).

> "The *pilot* follows the *teacher*. The *farmer* is helped by the *teacher*"

### 2.2.2 NON-EUCLIDEAN GRAPHS.

Some semantic domains are not exactly representable in an Euclidean space following a polar code. This occurs in two settings; (1) when relation composition is non-commutative (2) when relations are many-to-many or (3) when the underlying graph has non-Euclidean topology. Nevertheless, we ask whether a polar code can hold *locally*—within small neighborhoods—even when if not globally coherent.

**Family trees.** Then, we study a semantic domain with non-commutative and many-to-many relations: family trees. Parent relations compose non-commutatively (e.g., *mother ∘ father ≠ father ∘ mother*), and kinship ties such as *sibling of* are many-to-many (e.g., three sisters are each siblings of the others). Relational graphs are constructed where entities are people (e.g., *Joseph*, *Amelia*, *James*, *Emily*) and relations are kinship ties (e.g., mother of/father of; daughter of/son of; sister of/brother of). Valid relational graphs are sampled while enforcing classical genealogical constraints (no cycles through parent links, consistent parentage, and gendered inverse relations). For the "sibling of" relation, because of its many-to-many nature, we do not assign it to a prototypical direction; however, we include it in the structural loss computation.

> "*Amelia* is the mother of *James*. *Joseph* is the father of *James*.
> *Emily* is the sister of *James*"

**Metro maps.** Finally, we investigate a semantic domain where the graph's distance metric is not Euclidean. Metro networks are a canonical example, since the distance between two stops on different lines is determined by network path length rather than Euclidean geometry. Relational graphs are constructed where entities are metro stops corresponding to city landmarks (e.g., *lake*, *mall*, *hospital*) and relations denote the line and direction connecting adjacent stops (e.g., next on line A / previous on line A; next on line B / previous on line B). Transfer hubs occur where the lines intersect, and distances follow shortest-path length along the metro lines (including transfers).

> "The *mall* is one stop after the *lake* on line A.
> The *hospital* is one stop before the *mall* on line B"

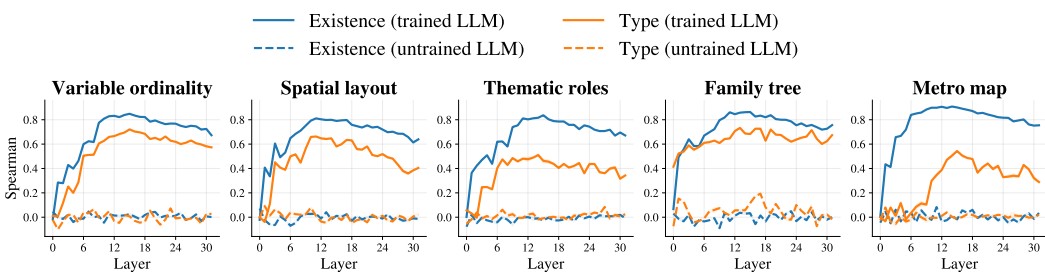

Figure 3: **Semantic structures are most linearly decodable in the middle layers, only in pretrained LLMs.** Spearman's $\rho$ for relation existence (blue) and type (orange) decoded by a polar probe from Llama3-8B across layers in five domains. In pretrained models (solid), decoding peaks around layers 12–15 and remains high in late layers. In randomly initialized models (dashed), both scores remain close to chance across all layers and domains.

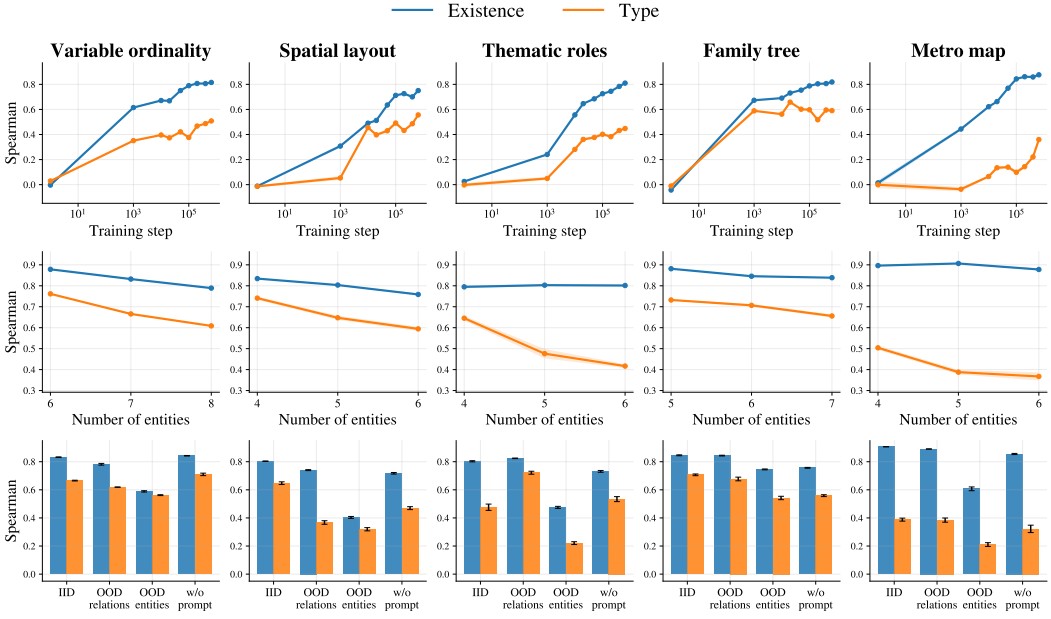

Figure 4: **Polar probe performance grows with pretraining, falls with the number of entities in the relational graph, and degrades with out-of-distribution (OOD) entities and relation surface forms. Top:** Spearman's $\rho$ for relation existence (blue) and type (orange) vs. pretraining steps at the best layer of OLMo-7B. **Middle:** Polar probe performance vs. number of entities in the graph at the best layer of Llama3.1-8B. **Bottom:** Generalization analysis on the best layer of Llama3.1-8B to OOD relation surface forms, OOD entity names, and without domain-specific prompting.

### 2.3 EVALUATION DETAILS

The test set for each dataset comprises 50 held-out graphs, each described 30 times. We report Spearman's $\rho$, following prior work (Hewitt & Manning, 2019), for (i) *relation existence* predictions (rank correlation between the vectorized upper triangle of predicted vs. gold pairwise distance matrices) and (ii) *relation type* predictions (rank correlation between the vectorized prototype-based predicted incidence tensor vs. the gold incidence tensor).

### 2.4 SUBSPACE ALIGNMENT SCORE.

Let $B_i \in \mathbb{R}^{k_i \times d}$ and $P_i \in \mathbb{R}^{k_i \times \mathcal{T}_i}$ be a trained polar probe and its prototype matrix (columns are prototype vectors). We project prototypes to model space:

$$V_i = B_i^{\dagger} P_i \in \mathbb{R}^{d \times \mathcal{T}_i},$$

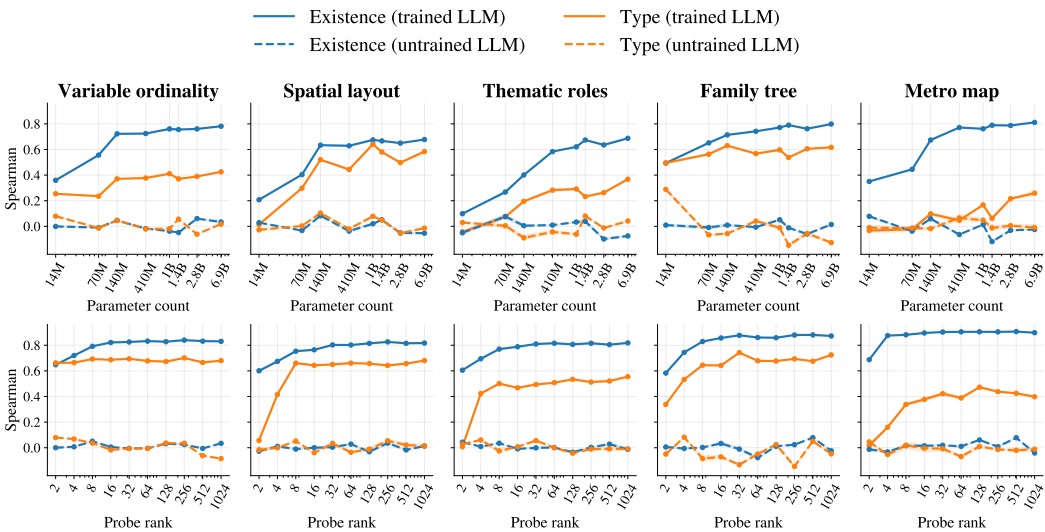

Figure 5: **Polar probe performance increases with model size and saturates at low probe rank. Top:** Spearman's $\rho$ for relation existence (blue) and type (orange) as a function of model size across the Pythia family, at the middle layer. Solid lines indicate pretrained models; dashed lines indicate randomly initialized ones. **Bottom:** Polar probe relation existence and type scores as a function of probe rank at the best layer of Llama3.1-8B.

and take the reduced QR factorization $V_i = Q_i R_i$ with $Q_i^\top Q_i = I_{\mathcal{T}_i}$, $Q_i \in \mathbb{R}^{d \times \mathcal{T}_i}$.

We report the mean squared cosines of the principal angles between two subspaces (Åke Björck & Golub, 1973):

$$\text{Alignment}(B_i, B_j) = \frac{1}{N} \left\| Q_i^\top Q_j \right\|_F^2 = \frac{1}{N} \sum_{n=1}^{N} \cos^2\theta_n, \qquad N = \min(\mathcal{T}_1, \mathcal{T}_2),$$

where $\{\theta_n\}$ are the principal angles. This score lies in $[0, 1]$, equals 0 for orthogonal subspaces, and increases with alignment.

## 2.5 CORRELATION WITH DOWNSTREAM PREDICTIONS.

We evaluate whether probe-space errors correlate with LLM downstream performance on a Question-Answering task. We present 500 relational graphs one-by-one to the LLM and, in each graph, query a specific relation (e.g., *Which variable is immediately greater than y?*). For each query, we record the logit assigned to the correct answer. To account for varying graph complexity, we normalize probe errors within each graph and report Spearman's $\rho$ between the normalized probe error and the correct answer's logit. A statistically significant negative correlation (Spearman's $\rho < 0$) would be consistent with larger probe errors being associated with lower logit values, suggesting that probe errors may provide a partial diagnostic of the model's reasoning performance.

## 2.6 PRETRAINED LARGE LANGUAGE MODELS.

We use publicly available, text-only LLMs. Most analyses are conducted with Llama3.1-8B [2] (Grattafiori et al., 2024). For the emergence analysis, we use OLMo-7B (Groeneveld et al., 2024), whose training checkpoints are publicly available[3]. For model-size analyses, we consider members of the Pythia[4] family (Biderman et al., 2023) with varying parameter counts ranging from 10 million to 6.8 billion.

---

[2]https://huggingface.co/meta-llama/Llama-3.1-8B

[3]https://huggingface.co/allenai/OLMo-7B

[4]https://huggingface.co/collections/EleutherAI/pythia-scaling-suite-64fb5dfa8c21ebb3db7ad2e1

## 3 RESULTS

**Layer analysis.** Across domains, polar probes decode semantic structures most accurately from the *middle* layers of Llama3-8B (Grattafiori et al., 2024) (figure 3). Spearman's $\rho$ for relation existence peaks at $\sim 0.80$ and for relation *type* at $\sim 0.50$–$0.70$ around layers 12–15. Probes trained on a randomly initialized Llama3-8B yield scores near 0.0 across layers, matching a random baseline. Unlike prior results in syntax (Hewitt & Manning, 2019; Diego-Simon et al., 2024), performance does not fully collapse in deeper layers. Domain-wise, type scores diverge: family trees reach comparatively high type accuracy already in shallow layers, likely reflecting lexical gender cues in names (encoded in the vocabulary embeddings), whereas metro maps show lower type scores overall and peak later, consistent with their non-Euclidean, network-based structure.

**Emergence during pretraining.** How does training shape their subspaces to represent semantic structures? To address this issue, we apply the Polar Probe to 9 checkpoints of OLMo-7B (Groeneveld et al., 2024). Polar probe performance increases with pretraining steps at the best-performing layer of OLMo-7B (figure 4). Scores for both relation existence and type strengthen gradually and remain largely unsaturated across available checkpoints, suggesting further gains with longer pretraining. Consistent with the depth analysis, decoding for metro maps emerges later during pretraining than for other domains and attains lower type scores overall. Overall, these results suggests that the polar coordinate principle is not a trivial property of high dimensional connectionist models, but directly depends on their ability to learn to store, represent, and manipulate knowledge.

**Graph complexity.** Are all semantic structures equally represented in the LLMs? To address this question we train and evaluate polar probes on relational graphs with varying number of entities. Both existence and type scores decline as the number of entities in the relational graph increases (measured at the best-performing layer of Llama3-8B) (figure 4). The drop is especially pronounced for relation *type* in thematic roles and metro maps, exceeding 50% when just two entities are added. As the entity count grows, the combinatorial space of possible graphs expands rapidly, making the decoding problem substantially harder.

**Generalization to new entities and relations.** While the above analyses are systematically evaluated on semantic structures absent from the training set, the Polar Probe may learn some relations by heart (e.g. plane left of ball = dimension 42). Consequently, we perform the same analyses on semantic structures, for which every entity name or relation surface form is absent from the training set. Figure 4 shows that, at the best-performing layer of Llama3-8B, polar probe performance degrades only modestly when the domain-specific prompt is removed. Using OOD relation surface forms produces an additional but modest drop. In contrast, OOD entity names have a substantially larger impact. Across all settings, polar probe performance remains well above the random baseline, which thus indicates that this polar coordinate system reliably generalizes to new structures.

**LLM size.** To evaluate whether the capacity of the LLMs influenced the geometry of semantic structures, we trained and evaluated polar probes on models from the Pythia suite spanning 14M–6.9B parameters (Biderman et al., 2023). Polar probe performance increases with model size, for each of the 5 domains, even thought the probe size is fixed at 128 (figure 5). This result is not trivially explained by the LLM dimensionality: when trained on randomly initialized LLMs, the polar probes remain close to chance across all sizes.

**Polar probe rank.** To assess whether the Polar Probe relies on dense or sparse representations, we trained and evaluated polar probes for each semantic domain with ranks logarithmically spaced from 2 to 1024.

Polar probe performance saturates at low ranks for both relation existence and type scores. Across domains, a rank of roughly 32 captures most of the achieved performance (figure 5). Notably, for variable ordinality, relation type scores peak with only 2 dimensions; beyond these values, increasing rank yields no consistent gains. Overall, this suggests that the semantic structures are represented in a compact subspace of the LLMs.

**Correlation with downstream predictions.** Are these polar representations epiphenomenal, or do they reflect the representations effectively used by the LLMs? To investigate this issue, we

perform a representation - behavior evaluation. Specifically, we compare the reliability of these representations (Polar Probe performance) to LLMs' ability to effectively answer questions about the semantic structure. We find that higher type errors in probe-space correlate with the worse Llama3-8B downstream performance, with the strongest correlation at layer 24 (figure 6). No analogous correlation is observed for existence errors.

**Subspace superposition.** Across domains, the learned subspaces are largely disjoint in the LLM's activation space (figure 7). A clear exception is a pronounced overlap between spatial layouts and variable ordinality, consistent with the superposition hypothesis (Elhage et al., 2022).

## 4 DISCUSSION

**Specific contributions.** This work demonstrates that the textual description of a new semantic structure can be recovered from the hidden activations of a large language model (LLM) with a simple geometric principle. Specifically, the *existence* and the *type* of a semantic relation between two entities are encoded by the *distance* and the *direction*, respectively, between their embeddings in a subspace of the LLM.

**Beyond syntactic trees.** Our approach extends earlier work on syntactic tree representations in LLMs (Hewitt & Manning, 2019; Diego-Simon et al., 2024), which focused exclusively on universal-dependency relations between words. By contrast, we here show that the Polar Probe is not limited to (1) syntax or to (2) to tree structures, but also extends to a broader class of structures, namely, directed and labeled Euclidean graphs.

**Beyond knowledge retrieval.** Linear probing of language models has been widely explored (Humphrey et al., 1970; Georgopoulos et al., 1986; Alain & Bengio, 2017; Conneau et al., 2018), to show that a broad range of features are linearly decodable from their activations: linguistic properties (Tenney et al., 2019; Jawahar et al., 2019; Liu et al., 2019), spatial and temporal knowledge (Gurnee & Tegmark, 2024; Chen et al., 2023), lexical semantics (Mikolov et al., 2013; Park et al., 2025b), political stances and factuality (Kim et al., 2025; Marks & Tegmark, 2024), and numeric values (Levy & Geva, 2025). These studies, however, primarily investigate how models retrieve knowledge acquired during training. By contrast, our present work targets compositional structures that, by design, could not be learned by heart during training. In this sense, this study closely relates to *in-context learning* (Brown et al., 2020; Park et al., 2025a) and binding (Feng & Steinhardt, 2024; Dai et al., 2024), providing a geometric principle for how compositional representations may be structured in neural activations.

**The limit of Euclidean graphs.** As noted in Simon et al. (2025), the Polar Probe relies on linear distances, and is therefore tailored to capture *Euclidean graphs*, i.e., structures whose nodes and edges can be faithfully embedded in a flat vector space and where distances satisfy Euclidean geometry. As a result, our method cannot accurately represent entire non-commutative, many-to-many relations and shortest-path distances (e.g. family trees and metro networks).Future work should therefore explore *non-Euclidean polar probes* (e.g., hyperbolic), which would use geodesic distance to encode the *existence* of relations and tangent-space directions to encode their *type*, extending the line of research initiated by Chen et al. (2021).

**The limit of correlational probing.** Our analyses reveal that the performance of the Polar Probe correlates with the LLM's ability to answer questions about the underlying semantic structure. However, this correlation does not imply that the model necessarily *uses* Polar Representations for its inference. To verify this hypothesis, targeted intervention experiments will be required (e.g., Nanda et al. (2023); Turner et al. (2025)).

**Larger impact.** Overall, this study clarifies how symbolic structures can be represented within vectorial systems, thereby refining the solution to the long-standing — and sometimes overstated — tension between symbolic and connectionist approaches to AI (Fodor & Pylyshyn, 1988; Smolensky, 1991; Marcus, 2003; Gayler, 2004).

**Reproducibility statement.** We describe all components needed to reproduce our results in the main text and appendix: dataset construction (§ 2.2, algorithm 1, table 1), probe objectives and hyperparameters (§ 2.1, § 2.1), evaluation metrics and procedures (§ 2.3). We list all the LLMs used in the experiments (§ 2.6), and these are publicly available. With the camera-ready, a public repository will be released containing data-generation, experiment, evaluation and plotting scripts together with exact package versions.

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

# 5 APPENDIX

## 5.1 CORRELATION WITH DOWNSTREAM PREDICTIONS

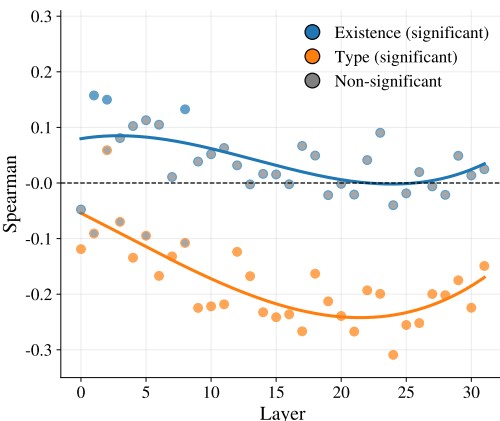

Figure 6: **Type probe errors predict LLM's downstream performance.** Layerwise Spearman correlation between existence (blue) and type (orange) probe-space errors and logit of the correct answer on a Question-Answering task over semantic structures.

## 5.2 SUBSPACE SUPERPOSITION

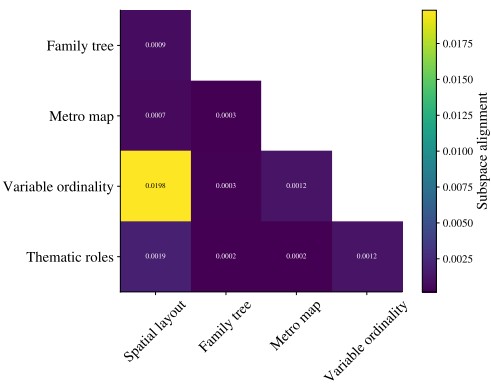

Figure 7: **Semantic domain subspaces are largely disjoint, with a spatial–ordinal overlap.** Cross-domain alignment at the best layer of Llama3-8B, quantified via the principal angles in LLM space (higher = more overlap).

## 5.3 LLM USAGE

LLM usage was limited to copy-editing, improving wording/clarity, enforcing style consistency, and minor LaTeX formatting.

## 5.4 DATASET GENERATION

---

**Algorithm 1** Grid-based Euclidean graph sampler

---

**Require:** entities $n$, dimension $d$
**Ensure:** $G = (\mathcal{V}_G, \mathcal{E}_G, \mathcal{T}_G)$
1: $\mathcal{V}_G = \{v_1, \ldots, v_n\}$
2: $\mathcal{T}_G = \{t_1, \ldots, t_d\}$
3: $U_d \leftarrow \{\mathbf{e}_1, \ldots, \mathbf{e}_d\}$           ▷ positive unit steps
4: positions $\leftarrow \{\mathbf{0}\}$           ▷ positions in $\mathbb{Z}^d$
5: **while** $|\text{positions}| < n$ **do**       ▷ random signed unit-step growth
6:    pick $\mathbf{x} \in$ positions uniformly
7:    pick $\mathbf{u} \in U_d$ uniformly
8:    pick $s \in \{-1, +1\}$ uniformly
9:    positions $\leftarrow$ positions $\cup \{\mathbf{x} + s\mathbf{u}\}$      ▷ reject if already occupied
10: **end while**
11: pick a bijection $f : \text{positions} \xrightarrow{\sim} \mathcal{V}_G$
12: pick a bijection $g : U_d \xrightarrow{\sim} \mathcal{T}_G$
13: $\mathcal{E}_G \leftarrow \varnothing$
14: **for all** ordered pairs $(v_i, v_j)$ **do**
15:    $\mathbf{\Delta} \leftarrow f^{-1}(v_j) - f^{-1}(v_i)$
16:    **if** $(\|\mathbf{\Delta}\|_2 = 1) \wedge (\mathbf{\Delta} \in U_d)$ **then**
17:      $t \leftarrow g^{-1}(\mathbf{\Delta}) \in \mathcal{T}_G$
18:      $\mathcal{E}_G \leftarrow \mathcal{E}_G \cup \{(v_i, v_j, t)\}$         ▷ Add to graph
19:    **end if**
20: **end for**
21: **return** $G = (\mathcal{V}_G, \mathcal{E}_G, \mathcal{T}_G)$

---

## 5.5 SETUP DETAILS

Table 1: Entity names, relations and prompt used for each semantic domain

| Spatial layout | Family tree | Metro lines | Vsriable ordinality | Thematic roles |
|---|---|---|---|---|
| **Entity names** | **Entity names** | **Entity names** | **Entity names** | **Entity names** |
| bag, ball, clock, lamp, ring, key, train, plane, boat, bus, bike, shirt, coin | James, Henry, Peter, Ben, Michael, Joseph, Isabel, Alice, Amelia, Charlotte, Isabella, Grace, Anna, | market, studio, pub, park, Opera, court, college, lake, airport, church, restaurant, hospital, mall | d, f, c, y, u, z, o, g, r, e, j, l, n | tourist, farmer, mechanic, scientist, teacher, manager, pilot, waiter, firefighter, student, doctor, engineer, actor |
| **Relations** | **Relations** | **Relations** | **Relations** | **Relations** |
| to the right of / to the left of, on top of / below | son of, daughter of, mom of, dad of, brother of, sister of | one stop before / one stop after | less than / greater than | follows / is followed by, helps / is helped by |
| **Entity names (OOD)** | **Entity names (OOD)** | **Entity names (OOD)** | **Entity names (OOD)** | **Entity names (OOD)** |
| car, cat, dog, tree, house, chair, table, book, door, bed, phone, cup, shoe | Victoria, Emily, Emma, Mary, Sarah, Lucy Leo, William, Paul, John, Bob, Mark, Daniel | bar, hill, square, pool, gallery, school, river, shop, tower, station, forum, bridge, library | w, b, h, x, i, m, v, t, s, k, q, a, p | driver, baker, dancer, neighbor, singer, librarian, lawyer, officer, artist, writer, chef, nurse, coach |
| **Relations (OOD)** | **Relations (OOD)** | **Relations (OOD)** | **Relations (OOD)** | **Relations (OOD)** |
| rightward of / leftward of, over / underneath | offspring of, mother of, father of, sibling of | one stop back from / one stop ahead of | fewer than / larger than | pursue / is pursued by, assists / is assisted by |
| **Prompt** | **Prompt** | **Prompt** | **Prompt** | **Prompt** |
| **Geometrical Layout** You will receive a description of an abstract scene laid out on a regular square grid. **Vocabulary** The scene contains objects, named: [] **Relations** The following spatial relations are used to describe the relative position of the objects on the grid: (above, below, left of, right of) **Goal** Infer the grid coordinates of every object so that **all** spatial constraints are satisfied. **Scene description:** | **Family Tree** You will receive a description of a family tree. **Vocabulary** The family tree contains people, named: [] **Relations** The following family relations are used to describe link between two people in the family: (dad of, mom of, son of, daughter of, brother of and sister of) **Goal** Infer the family relations among all members in the family so that **all** constraints are satisfied. **Scene description:** | **Metro Map** You will receive a description of a city's metro map. **Vocabulary** The metro can have lines [] which connect the following sites in the city []. **Relations** The following relations are used to describe where each site is located on a given direction of the metro line: (one stop before, one stop after) **Goal** Infer the position of each site on its corresponding metro line so that **all** constraints are satisfied. **Scene description:** | **Mathematical Variables** You will receive a description of set of mathematical constants, each associated with an unknown real number. **Vocabulary** The constants are named according to []. **Relations** The following relations are used to describe the relative position of a constant with respect to another one on the real axis: (greater than, less than) **Goal** Infer the position of each constant on the real axis so that **all** constraints are satisfied. **Scene description:** | **Social Interactions** You will receive a description of an interaction between several people. **Vocabulary** The people potentially participating in the interaction are: [] **Relations** The interaction type between two people can be: (follow - be followed by , help - be helped by) **Goal** Infer the interactions among participants so that **all** constraints are satisfied. **Scene description:** |