# OpenReview forum: "Polar probe linearly decodes semantic structures from LLMs"
_ICLR.cc/2026/Conference — Submitted to ICLR 2026_

### Official Review · Reviewer_FrSx · 2025-10-29

**Soundness:** 3
**Presentation:** 2
**Contribution:** 1
**Rating:** 2
**Confidence:** 4

**Summary:**

This paper investigates how Large Language Models (LLMs) represent and bind concepts to form complex semantic structures provided in-context. The central hypothesis is that LLMs utilize a specific geometric code: in a low-dimensional subspace, the existence of a semantic relation is encoded by Euclidean distance, and the type of relation is encoded by direction (angle). The authors test this by training a "Polar Probe" on LLM activations from five different minimalist graph domains (e.g., spatial layouts, family trees). The findings suggest that this structure is linearly decodable, emerges in the middle layers, and correlates with downstream QA performance.

**Strengths:**

Methodical Experimental Design: The paper is structured methodically. The central hypothesis is clear, and the authors test it with a reasonable set of ablations (model size, layer, graph complexity, OOD generalization).

Focus on In-Context Binding: The paper correctly identifies an important area of research: understanding how LLMs build new representations on the fly, rather than just probing for static, pre-trained knowledge.

Link to Downstream Function: The inclusion of a correlation analysis (Figure 6) between probe error and QA performance is a necessary and welcome step, providing preliminary evidence that these representations are not merely epiphenomenal.

**Weaknesses:**

1. Highly Incremental Novelty: The paper’s central weakness is its significant lack of novelty. The core methodology is a direct application of an existing tool (the Polar Probe) to a new set of synthetic datasets. The paper’s central finding — that (spatial) relations are encoded geometrically by direction — is not new. For example, similar work (Tehenan et al., "Linear Spatial World Models Emerge in Large Language Models"), has already demonstrated this finding and even provided stronger, causal evidence using interventions. The authors’ contribution thus appears to be merely an incremental validation of a known hypothesis on a few new toy domains.

2. Failure to Scale (Minimalist Tasks): The experiments are confined to “minimalist” semantic structures (e.g., graphs with 4–6 entities). The paper’s own results (Figure 4, middle) show severe performance degradation as the number of entities increases. This is a critical flaw, as it strongly suggests that the “simple geometrical principle” the authors claim to have found does not scale and is an artifact of the toy-like simplicity of the tasks. This makes the findings largely non-generalizable to the complex, large-scale semantic graphs found in real-world text.

3. Fundamentally Correlational: The paper is a classic “probing” study. It shows that a linear decoder can be trained to extract information, but it provides no causal evidence that the model actually uses this geometric structure to perform computations. The link to QA (Figure 6) is itself merely correlational. The interpretability community has largely moved beyond such correlational analyses, and the lack of any interventional experiments (e.g., activation steering) makes the paper’s claims feel weak and dated.

4. Thin Content and Weak Visualizations: The paper feels “padded.” The Results section is largely a textual description of the figures, with little substantive analysis.

**Questions:**

1. The core finding — that a geometric code exists for in-context spatial relations — has been shown in prior work, often with causal evidence. Given this, what is the specific, novel contribution of this paper beyond applying an existing probe to new, minimalist datasets?

2. The performance collapse shown in Figure 4 (middle) is alarming. How can the authors claim this represents a “principle” of semantic binding if it fails on graphs with just six entities? Does this not suggest that the “polar code” is a brittle artifact of “toy” problems?

3. The correlation with downstream performance (Figure 6) is the paper’s most compelling result. However, can you explain the discrepancy — why does type (angular) error correlate with task performance, while existence (distance) error does not?

4. Why did you choose to visualize the subspace using PCA in Figure 2 rather than by projecting the relation vectors onto the learned prototype vectors, which would provide a more direct evaluation of the central hypothesis? If the PCA plot conveys some meaningful structure, what do PC1 and PC2 represent?

5. You use Spearman’s rank correlation for all analyses. Could you provide a plot of the original values used for the correlation? I’m not sure why Spearman’s rank correlation was chosen in every case — perhaps another metric could be more appropriate.

6. Could you provide any baseline model to justify why you use a polar probe? A comparison to a mean-difference or linear probe would be interesting.

7. Do you foresee any way this polar probe could be used to actively steer or manipulate LLM behaviors?

---

> ### Author Response · Authors · 2025-11-20
>
> We thank Reviewer FrSx for their sharp and detailed comments and hope our response will clarify the points raised.
>
> &nbsp;
>
> We provide an anonymized URL to additional analyses including causal interventions (Figure 8 and Figure 9) and naturalistic evaluations (Figure 14):
>
> https://drive.google.com/file/d/14Ft-qaBAHjZjAhXnkHq-odFR-8fPJn7H/view?usp=sharing
>
> ---
>
> * > The paper’s central weakness is its significant lack of novelty.
>
>    We extend the probing framework to compositional semantics. To the best of our knowledge no other study has used this framework to model semantic structures. Additionally, our controlled setting deviates from (Hewitt & Manning, 2019 ; Diego-Simon, 2024\) where probes were trained over naturalistic corpora without controlling for surface confounds.
>
>    &nbsp;
>
>    To strengthen the novelty of our work, we have now extended our results by performing causal interventions with polar probes.
>    The result for this analysis is available at the following anonymized URL
>
>    (Figure 8 and Figure 9): https://drive.google.com/file/d/14Ft-qaBAHjZjAhXnkHq-odFR-8fPJn7H/view?usp=sharing
>
> ---
>
> * > The paper’s central finding — that (spatial) relations are encoded geometrically by direction — is not new. For example, similar work (Tehenan et al., "Linear Spatial World Models Emerge in Large Language Models"), has already demonstrated this finding and even provided stronger, causal evidence using interventions.
>
>    We thank the reviewer for pointing out (Tehenan, 2025), we were not aware of this concurrent work which we have added to the Related Work section.
>
>    &nbsp;
>
>    (Tehenan, 2025\) differs from our contribution in the following aspects:
>
>    1. **Broader scope:** Here we go far beyond modeling 2D or 3D spatial coordinates: and show the viability (and limits) of polar probes on conceptual (and non-euclidean) semantic domains such as Thematic roles, Family trees and Metro maps.
>    2. **Methodological difference:** (Tehenan, 2025\) regresses model activations onto ground-truth 2D or 3D coordinates similarily to (Gurnee, 2024). This approach cannot be generalized to conceptual semantic graphs, where there exists no absolute ground-truth coordinates. Conceptual graphs contain only relative relational structure, which in turn, can be captured by polar probes.
>
> ---
>
> * > The experiments are confined to “minimalist” semantic structures (e.g., graphs with 4–6 entities). The paper’s own results (Figure 4, middle) show severe performance degradation as the number of entities increases.
>
>   LLMs are known to face challenges on compositional and reasoning tasks (Benchekroun, 2023  Figure 6 ; Dziri, 2023 ; Shojaee, 2025). In our experiments, computational constraints limit us to using 8B models without chain-of-thought reasoning. As a result, probe performance is inherently capped by the LLM’s downstream competence. Extending this analysis to larger reasoning models, and examining how such geometric structures develop throughout the reasoning trace, constitutes a promising avenue for future work.
>
>    &nbsp;
>
>   We clarify this point in the Discussion section:
>   ```
>   Our study is limited to 8B models without chain-of-thought reasoning, which constrains the model’s downstream performance and therefore the attainable probe accuracy. Examining larger reasoning models could reveal how semantic structures emerge and evolve throughout their reasoning traces, presenting an exciting direction for future work.
>    ```
>
>    &nbsp;
>
>    We also clarify that the y-axis in Figure 4, middle panel starts at 0.3. Possibly misleading the plot's interpretation. We change the plot to avoid confusion.
>
>    &nbsp;
>
>    Finally, we have created two LLM-generated naturalistic datasets (English and multilingual) for Spatial Layouts. The results show that Polar probes generalise to naturalistic and multilingual sentences. Existence and Type scores substantially exceed both chance level and the untrained baseline.
>
>    The result is available at (Figure 14) in the following anonymized URL:
>
>    https://drive.google.com/file/d/14Ft-qaBAHjZjAhXnkHq-odFR-8fPJn7H/view?usp=sharing
>
>
> ---
>
> * > the lack of any interventional experiments (e.g., activation steering) makes the paper’s claims feel weak and dated
>
>   We agree with the reviewer that correlational analyses alone cannot establish whether the geometric structure we decode is functionally relevant for the model. Demonstrating causal manipulations is indeed an important direction of work.
>
>    &nbsp;
>
>   To address this point, we conducted activation steering experiments using the polar-probe prototype vectors.
>
>   Our results indicate that such interventions do influence the model’s answers in the QA task, especially in the middle layers, where the probes achieve their best decoding performance.
>
>    &nbsp;
>
>   This result is available at the following anonymized URL:
>
>   (Figure 8 and Figure 9\) https://drive.google.com/file/d/14Ft-qaBAHjZjAhXnkHq-odFR-8fPJn7H/view?usp=sharing

---

> ### Author Response · Authors · 2025-11-20
>
> * > Thin Content and Weak Visualizations: The paper feels “padded.” The Results section is largely a textual description of the figures, with little substantive analysis.
>
>   We are happy to extend the results section in the extra page of the camera-ready version of the paper.
>
>    As for the visualizations, we are also willing to change them, can you be more specific about what could be improved?
>
> ---
>
> **Questions**
>
> * > The correlation with downstream performance (Figure 6) is the paper’s most compelling result. However, can you explain the discrepancy — why does type (angular) error correlate with task performance, while existence (distance) error does not?
>
>    We believe this difference in correlation scores is due to the nature of the QA task. An example of a question in the QA dataset would be; “What variable is immediately greater than x?” The task does not involve inferring how far in the graph two variables are in the semantic graph but rather whether or not they are connected and the type of relation. An alternative task asking “How many variables are there in between x and z?” may result in a stronger distance correlation.
>
> ---
>
> * > Why did you choose to visualize the subspace using PCA in Figure 2 rather than by projecting the relation vectors onto the learned prototype vectors, which would provide a more direct evaluation of the central hypothesis? If the PCA plot conveys some meaningful structure, what do PC1 and PC2 represent?
>
>    Projecting onto the prototypes results in a similar figure, however, because PCA components are orthoghonal, projections look better. Because polar prototypes are learned, such vectors might not be exactly orthogonal.
>
> ---
>
> * >You use Spearman’s rank correlation for all analyses. Could you provide a plot of the original values used for the correlation? I’m not sure why Spearman’s rank correlation was chosen in every case — perhaps another metric could be more appropriate.
>
>    Spearman correlation was used for consistency with previous work on Structural Probes. You can find the requested figure (pred vs. true) at the following anonymized link.
>
>    (Figure 12\) https://drive.google.com/file/d/14Ft-qaBAHjZjAhXnkHq-odFR-8fPJn7H/view?usp=sharing
>
> ---
>
> * > Could you provide any baseline model to justify why you use a polar probe? A comparison to a mean-difference or linear probe would be interesting.
>
>    Good idea, we have now added both baselines, these highlight the advantage of polar probes in decoding semantic structure from LLM activations.
>
>    (Figure 13\) https://drive.google.com/file/d/14Ft-qaBAHjZjAhXnkHq-odFR-8fPJn7H/view?usp=sharing
>
>    &nbsp;
>
>    The baselines added are:
>
>    1. **No-probe, just trained prototypes**: shows the importance of learning a probe rather than considering raw LLM activations
>    2. **Linear baseline**: measures how well linear distances predict ground-truth semantic distances
>
> ---
>
> We hope our response  and additional analyses clarified adequately the reviewer’s concerns and, if so, that this is reflected in the updated score.
>
> ---
>
> **Citations**
>
> (Gurnee, 2023\) https://arxiv.org/pdf/2310.02207
> (Hewitt & Liang, 2019\) https://aclanthology.org/D19-1275.pdf
> (Maudslay, 2021\) https://aclanthology.org/2021.naacl-main.11.pdf
> (Belinkov, 2022\) https://aclanthology.org/2022.cl-1.7.pdf
> (Lake & Baroni, 2018) https://arxiv.org/pdf/1711.00350
> (Hupkes, 2020\) https://arxiv.org/pdf/1908.08351
> (Lakretz, 2022\) https://aclanthology.org/2022.coling-1.285.pdf
> (Benchekroun, 2023\) https://arxiv.org/pdf/2311.15930
> (Dziri, 2023\) https://arxiv.org/pdf/2305.18654
> (Shojaee, 2025\) https://arxiv.org/pdf/2506.06941

---

> > ### Comment · Reviewer_FrSx · 2025-11-27
> > **Response to Authors**
> >
> > Thank you for your clarification! I appreciate the inclusion of the causal intervention experiment. However, I still have some remaining questions:
> > 1. Why is "normalized probability" shown in Figure 8? I am interested in the results for the real (raw) probability. Can the target tokens actually become the Top-1 prediction? Also, why does a large steering parameter worsen performance?
> > 2. Could you show the performance when the number of entities is larger than 6 (e.g., 10 or 20)?
> > 3. The paper's narrative structure is still confusing. For example, Section 2.4 appears abruptly, while it seems related to the results in the Appendix. The "punch line" is hard to grasp. I strongly suggest moving standard experimental details to the Appendix. For the Results section, the main confusion arises because the figures are too far from the text. I suggest selecting just two representative datasets (e.g., Spatial Layout and Family Tree) for the main text and presenting them in a matrix format (datasets as rows, analyses as columns). This would allow you to easily add the additional experiments and help readers grasp the key points at a glance by keeping the figures and text together. The results for the other datasets can be moved to the Appendix. Please reconsider this suggestion.
> > 4. Have you actually conducted the experiment on the alternative task asking “How many variables are there in between x and z?”
> > 5. What happens when you visualize the projection onto the prototypes? Can you show that result? Orthogonality in the PCA plot does not imply orthogonality in the full dimensions. Why do you think the PCA plot demonstrates such orthogonality?

---

> > > ### Author Response · Authors · 2025-12-01
> > >
> > > We thank the reviewer for the constructive exchange and for their encouraging remarks on the causal intervention experiments.
> > >
> > > The discussion was productive and led to a few remaining questions, which we address below.
> > >
> > >    &nbsp;
> > >
> > > * > Why is "normalized probability" shown in Figure 8? I am interested in the results for the real (raw) probability. Can the target tokens actually become the Top-1 prediction? Also, why does a large steering parameter worsen performance?
> > >
> > >    - We have now added a new figure with the raw probability, the results remain consistent. Normalised probability reflects the relative probability of the correct token with respect to all other tokens present in the semantic structure.
> > >    - The reason for this is to ensure our result is not the result of uninterseting phenomena eg: all entity names equally increase in probability.
> > >    - With steering, target tokens do not become top-1, but they do increase in probability significantly.
> > >    - Performance begins to degrade at higher steering values (≈12). This is consistent with the polar hypothesis: strong steering increases the representation’s norm, enlarging graph distances and lowering contiguity probabilities.
> > >
> > > * > Could you show the performance when the number of entities is larger than 6 (e.g., 10 or 20)?
> > >
> > >    - Performance on large semantic graphs is low for smaller models, as supported in our author response. We agree that a more detailed analysis of the scaling is informative and will include it in the camera-ready version.
> > >
> > > * > The paper's narrative structure is still confusing. For example, Section 2.4 appears abruptly, while it seems related to the results in the Appendix. The "punch line" is hard to grasp. I strongly suggest moving standard experimental details to the Appendix. For the Results section, the main confusion arises because the figures are too far from the text. I suggest selecting just two representative datasets (e.g., Spatial Layout and Family Tree) for the main text and presenting them in a matrix format (datasets as rows, analyses as columns). This would allow you to easily add the additional experiments and help readers grasp the key points at a glance by keeping the figures and text together. The results for the other datasets can be moved to the Appendix. Please reconsider this suggestion.
> > >
> > >    - We thank the reviewer for these editorial suggestions, they are very helpful. We have reorganized the main figures and moved some content to the Appendix for the camera-ready version.
> > >
> > > * > Have you actually conducted the experiment on the alternative task asking “How many variables are there in between x and z?”
> > >
> > >    - Our current focus is on the causal intervention experiments, but we agree that this complementary correlational analysis is valuable and will include it in the camera-ready version.
> > >
> > > * > What happens when you visualize the projection onto the prototypes? Can you show that result? Orthogonality in the PCA plot does not imply orthogonality in the full dimensions. Why do you think the PCA plot demonstrates such orthogonality?
> > >
> > >    - We have added a figure showing the projection onto prototypes.
> > >    - We clarify on what these projections mean:
> > >      - Projecting onto prototypes shows the performance of the probe at telling the **type of relation between 2 directly-connected entities**. Thus, we do not expect that projecting onto the prototypes explains the semantic structure, since the radial aspect of the polar probe is not considered, only the angular one.
> > >       - Prototypes do not need to be orthogonal as they are learned jointly with the probe's parameters. When performing PCA, the components are orthogonal by definition, resulting in a cleaner plot.
> > >
> > > We provide an anonymised link to the additional results mentioned above:
> > >
> > > https://drive.google.com/file/d/1g9StirFjm6NPJm-VB6ZZGeQQ-aF-Z_m-/view?usp=sharing
> > >
> > > ---
> > >
> > > We hope our responses and the additional analyses help clarify these points.

---

### Official Review · Reviewer_NUp5 · 2025-10-31

**Soundness:** 3
**Presentation:** 3
**Contribution:** 2
**Rating:** 6
**Confidence:** 3

**Summary:**

This paper applies the polar probe method for probing for the existence and direction of syntactic relationships to semantic structures. Their core result is showing that polar probes can indeed decode both the existence and type of semantic relationships such as those in arithmetic statements, family trees, etc, just as they can for syntactic relationships. The paper analyzes these probes - finding they emerge over training, are more effective at middle layers, saturate at low dimensions, degrade with more complex relationships and improve with model scale.

**Strengths:**

The authors are careful in their application of the polar probe technique from Diego-Simon 2024, and study a wide variety of semantic relationships across a wide variety of model families and sizes and even perform analysis across model checkpoints.

**Weaknesses:**

I guess I feel that the core results of this paper are not surprising. The authors cite Park 2025 (in-context learning of representations) which shows that contextualized embeddings of tokens arranged along the vertices of a graph structure sampled by a random walk will capture the geometry of that graph structure, which seems to largely predict the results of this paper. There's other examples in the literature of probing for semantic relationships e.g. https://arxiv.org/abs/2406.01506, https://aclanthology.org/2021.acl-long.36/ or https://aclanthology.org/2021.acl-long.145/

**Questions:**

I wonder whether the authors studied uncontextualized embeddings for tokens with pre-built semantic structure (e.g. "father" "son" "mother" etc or numbers)?

---

> ### Author Response · Authors · 2025-11-20
>
> We thank reviewer NUp5 for their constructive feedback and hope our response will clarify the points raised.
>
> &nbsp;
>
> We provide an anonymized URL to additional analyses including causal interventions (Figure 8 and Figure 9) and naturalistic evaluations (Figure 14):
>
> https://drive.google.com/file/d/14Ft-qaBAHjZjAhXnkHq-odFR-8fPJn7H/view?usp=sharing
>
>
> ---
>
> * > The authors cite Park 2025 (in-context learning of representations) which shows that contextualized embeddings of     tokens arranged along the vertices of a graph structure sampled by a random walk will capture the geometry of that graph structure, which seems to largely predict the results of this paper.
>
>   We agree with the reviewer that Park (2024) is closely related to our work and that the distinctions between the two were not made sufficiently clear. We clarify this important distinction in the Discussion section:
>
>   ```
>   By contrast, our present work targets compositional structures that, by design, could not be memorized during training. In this sense, our study is related to in-context learning (Brown et al., 2020; Park et al., 2025a), but we do not rely on statistical learning of in-context regularities. Instead, we investigate how LLMs recover semantic structure by binding entities to known relations acquired during pretraining. Prior work has examined binding (Feng & Steinhardt, 2024; Dai et al., 2024); we extend this line of research by providing a geometric principle for how compositional representations may be organized in neural activations.
>
>   ```
>
>    &nbsp;
>
>    There are at least two important differences with our work.
>
>    1. **We do not train probes on random walks over semantic structures**
>
>         (Park 2024\) exposes the model to *random-walk sequences* that repeatedly traverse the underlying structure in-context. In contrast, we provide only a **single natural-language description** per semantic graph, so the model cannot reconstruct the relational structure from in-context transition statistics.
>
>     2. **We study semantic binding, not statistical in-context learning**
>
>        Our experiments test how LLMs achieve semantic binding, that is, parsing a semantic structure from a single description like “Bob is Alice’s father’’. Our findings reflect how LLMs understand language. By contrast, (Park 2024\) does not demonstrate linguistic processing, but shows that models learn *transition probabilities* between tokens to account for in-context.
>
> ---
> * > There's other examples in the literature of probing for semantic relationships
>
>   We thank the reviewer for pointing us to these works. We include them to the Related Work section.
>
>   &nbsp;
>
>      https://arxiv.org/abs/2406.01506 and https://aclanthology.org/2021.acl-long.36/ focus on lexical semantics, this differs from our study in that Lexical Semantics is about uncontextualized meaning, whereas we focus in binding and compositional semantics. We include these to the “*Related Work”* section.
>
>   &nbsp;
>
>   https://aclanthology.org/2021.acl-long.145/ also differs from our work in several aspects:
>
>   * They focus on naturalistic syntax and Abstract Meaning Representation (AMR).
>   * They do not analyze emergence across layers, model scale, or training checkpoints.
>   * They employ an information-theoretic probe, while we evaluate a specific geometric principle of representation.
>
> ---
>
> **Questions**
>
> * > I wonder whether the authors studied uncontextualized embeddings for tokens with pre-built semantic structure (e.g. "father" "son" "mother" etc or numbers)?
>
>   Great idea\!
>   We have now replicated the subspace-alignment analysis (Figure 7 in the Appendix) by adding uncontextualized embeddings such as “father,” “right,” “next,” “many,” and “follower,” one for each semantic domain.
>
>   &nbsp;
>
>   The result for this analysis is available at the following anonymized URL:
>
>   (Figure 11\) https://drive.google.com/file/d/14Ft-qaBAHjZjAhXnkHq-odFR-8fPJn7H/view?usp=sharing
>
>
> ---
>
> We hope our response  and additional analyses clarified adequately the reviewer’s concerns and, if so, that this is reflected in the updated score.

---

### Official Review · Reviewer_sDGr · 2025-11-01

**Soundness:** 2
**Presentation:** 3
**Contribution:** 3
**Rating:** 6
**Confidence:** 3

**Summary:**

The paper aims to rigorously exam a precise geometric hypothesis about LLM hidden states: after a linear projection, relation existence should be recoverable from radial distance (near vs. far), while relation type should align with angular direction in the same subspace. The study evaluates this kind of “polar” code across several controlled, synthetic domains designed to instantiate different forms of semantic structure—such as ordered sets, spatial layouts, role–filler templates, and small graph families including kinship- and metro-like relations. The empirical results indicate that decoding performance typically peaks in middle layers, improves with model size and pretraining progress, exhibits some out-of-distribution generalization (e.g., to novel entities or relation surface forms), and shows a negative correlation between probe error and simple downstream QA behavior. The claims are appropriately scoped to linear decodability rather than causal use by the model.

**Strengths:**

I like how precise the core hypothesis in the paper is: existence = distance, type = direction. And I think keeping the probe strictly linear helps me trust the interpretation. The evaluation protocol/setups also look correct to my eyes: the same geometric test across multiple semantic domains strengthens the result. I also like the emergence analyses (layer position, scaling etc.). The OOD checks and the correlation with QA, while not causal, are sensible sanity checks that the decoded geometry aligns with behavior. Overall, to the best of my knowledge, the paper is careful in its claims and matches them to what the experiments can actually support and are novel in terms of experiments construction for the hypothesis.

**Weaknesses:**

One potential drawback is the reliance on synthetic prompts. This leaves open whether the same distance–direction code appears in natural text, where relations are implicit and surface forms vary.  It would strenghthen author's arguments much more if they can construct or source some evaluation protocol based on natural/free texts. Without a small naturalistic check, it is uncertain how the polar code manifests in less controlled contexts.

**Questions:**

I have a curious question about where might euclidean geometry break: on the domains the authors find hardest, do failures concentrate in radial (existence) or angular (type) terms, and do they correlate with graph features (degree, cycles, path length)?

---

> ### Author Response · Authors · 2025-11-20
>
> We thank reviewer sDGr for their precise and helpful feedback and hope our response will clarify the points raised.
>
> &nbsp;
>
> We provide an anonymized URL to additional analyses including causal interventions (Figure 8 and Figure 9\) as well as more naturalistic evaluations (Figure 14\) :
>
> https://drive.google.com/file/d/14Ft-qaBAHjZjAhXnkHq-odFR-8fPJn7H/view?usp=sharing
>
> ---
> * > One potential drawback is the reliance on synthetic prompts
>
>    We agree that the reviewer that extending the evaluation of probes beyond controlled settings is an important direction.
>
>   We have now created two LLM-generated naturalistic datasets (English and multilingual) for Spatial Layouts. The results show that Polar probes generalise to these naturalistic and multilingual sentences.
>
>   Both Existence and Type scores substantially exceed chance level and the untrained baseline.
>
>   &nbsp;
>
>
>    The result is available at (Figure 14) in the following anonymized URL:
>
>    https://drive.google.com/file/d/14Ft-qaBAHjZjAhXnkHq-odFR-8fPJn7H/view?usp=sharing
>
>   &nbsp;
>
>    Furthermore, we acknowledge that the motivation for using controlled datasets and its resulting limitations were not sufficiently justified in the manuscript.
>
>
>   &nbsp;
>
>   We extend the Datasets section with the following paragraph:
>
>   ```
>   Probes are shown to be especially sensitive to spurious correlations and confounds present in training data (Hewitt & Liang, 2019; Maudslay, 2021; Belinkov, 2022). Such constraint motivates the use of controlled datasets, a methodological choice widely adopted in interpretability and compositionality studies (Lake & Baroni, 2018;  Hupkes, 2020; Lakretz, 2022).
>   ```
>
>   We extend the Discussion section with the following paragraph:
>
>   ```
>   We use synthetic, controlled datasets to eliminate surface confounds such as word-frequency or positional cues. This methodological choice enforces probes to not rely on spurious surface patterns and makes their evaluation more interpretable. However, this limits their direct applicability to naturalistic text, where relations might be implicit and surface forms vary widely.
>   ```
>
> ---
>
> **Questions**
>
> * > where might euclidean geometry break? ...  do [failures] correlate with graph features?
>
>   Great question,
>
>   &nbsp;
>
>   The failures can be compared by comparing the Type and Existence Spearman correlations for each semantic domain. Type scores are generally lower than those for Existence, meaning that failures concentrate in the “angular” part of the probe.
>
>   &nbsp;
>
>   We have also analyzed how radial errors vary as a function of the distance in the semantic graph. The result for this analysis is available at the following anonymized URL
>
>   &nbsp;
>
>   (Figure 10) https://drive.google.com/file/d/14Ft-qaBAHjZjAhXnkHq-odFR-8fPJn7H/view?usp=sharing
>
> ---
> We hope our response  and additional analyses clarified adequately the reviewer’s concerns and, if so, that this is reflected in the updated score.
>
> ---
>
> **Citations:**
> (Hewitt & Liang, 2019\) https://aclanthology.org/D19-1275.pdf
> (Maudslay, 2021\) https://aclanthology.org/2021.naacl-main.11.pdf
> (Belinkov, 2022\) https://aclanthology.org/2022.cl-1.7.pdf
> (Lake & Baroni, 2018) https://arxiv.org/pdf/1711.00350
> (Hupkes, 2020\) https://arxiv.org/pdf/1908.08351
> (Lakretz, 2022\) https://aclanthology.org/2022.coling-1.285.pdf

---

### Official Review · Reviewer_e9gD · 2025-11-02

**Soundness:** 3
**Presentation:** 4
**Contribution:** 2
**Rating:** 6
**Confidence:** 3

**Summary:**

The paper aims to understand how LLMs represent complex semantic structures. The authors propose polar probes to decode these structures from LLMs. A polar probe learns a linear mapping from an entity’s hidden state (activation) to a more interpretable subspace in which smaller distances between entities indicate the existence of a relation and the direction between entities indicates the relation type. To test whether polar probes capture semantic structure, the authors introduce tasks across five domains. The results show that semantic structure can be linearly recovered from LLMs’ activations. They also analyze when performance degrades and find that the quality of the polar representations is highly correlated with LLMs’ ability to answer questions.

**Strengths:**

- The paper is well written. Although organized slightly unconventionally, it is easy to follow and read.
- The key question, “How do artificial neural networks bind concepts to form complex semantic structures?”, is addressed through simple but precise experiments.
- The experiments show that polar probes are a useful tool for understanding semantics in LLMs, especially for Euclidean graph structures.

**Weaknesses:**

**Contributions.**
The main contribution is extending polar probes [a] from understanding syntactic structures to semantic structures. The toy datasets proposed in this work are a reasonable contribution but might not be sufficient on their own.

**Non-Euclidean graphs.**
While the performance of polar probes is above the random baseline, it is unclear whether it is reliable for non-Euclidean structures. On the metro-map dataset, type performance is close to the random baseline even with larger models. Adding another dataset with non-Euclidean structure would strengthen the claim that polar probes can decode semantic structure for non-Euclidean graphs as well.

**Minor.**
This is a minor comment/suggestion. The paper does not sufficiently motivate that there is a long-standing debate over whether neural networks or connectionist models can understand semantics. At the end of page 9, the authors mention this tension between symbolic and connectionist approaches, but it could be addressed much earlier; doing so would make the paper stronger, especially for a broader audience.

**Questions:**

See weaknesses.

---

> ### Comment · Reviewer_e9gD · 2025-11-12
> **References**
>
> **References**
>
> [a] A polar coordinate system represents syntax in large language models. NeurIPS 2024.
>
> (Missed including this in the main review.)

---

> ### Author Response · Authors · 2025-11-20
>
> We thank reviewer e9gD for their constructive feedback and hope our responses help clarify the points raised
>
> &nbsp;
>
> We provide an anonymized URL to additional analyses including causal interventions as well as more naturalistic evaluations:
>
> https://drive.google.com/file/d/14Ft-qaBAHjZjAhXnkHq-odFR-8fPJn7H/view?usp=sharing
>
> ---
>
> * > The toy datasets proposed in this work are a reasonable contribution but might not be sufficient on their own.
>
>     We agree that the reviewer that extending the evaluation of probes beyond controlled settings is an important direction. Therefore, we have created two LLM-generated naturalistic datasets (English and multilingual) for Spatial Layouts. The results show that Polar probes generalise to naturalistic and multilingual sentences. Both Existence and Type scores substantially exceed chance level and the untrained baseline.
>
>    &nbsp;
>
>     The result is available at (Figure 14\) in the following anonymized URL:
>
>    https://drive.google.com/file/d/14Ft-qaBAHjZjAhXnkHq-odFR-8fPJn7H/view?usp=sharing
>
>     &nbsp;
>
>     Furthermore, we acknowledge that the motivation for using controlled datasets and its resulting limitations were not sufficiently justified in the manuscript.
>
>    We clarify extending the Datasets section with the following paragraph:
>
>     ```
>     Probes are shown to be especially sensitive to spurious correlations and confounds present in training data (Hewitt & Liang, 2019; Maudslay, 2021; Belinkov, 2022). Such constraint motivates the use of controlled datasets, a methodological choice widely adopted in interpretability and compositionality studies (Lake & Baroni, 2018;  Hupkes, 2020; Lakretz, 2022).
>     ```
>
>     We also clarify by extending the Discussion section with the following paragraph:
>
>     ```
>     We use synthetic, controlled datasets to eliminate surface confounds such as word-frequency or positional cues. This methodological choice enforces probes to not rely on spurious surface patterns and makes their evaluation more interpretable. However, this limits their direct applicability to naturalistic text, where relations might be implicit and surface forms vary widely.
>     ```
>
>     Finally, to extend the paper’s contribution, we have performed causal interventions, successfully steering Llama3-8B with polar probes, the results are available at (Figures 8 and 9\) in the following anonymized URL:
>
>    https://drive.google.com/file/d/14Ft-qaBAHjZjAhXnkHq-odFR-8fPJn7H/view?usp=sharing
>
> ---
>
> * > On the metro-map dataset, type performance is close to the random baseline even with larger models.
>
>   In its current form, the polar probe finds first-order approximations for non-euclidean graphs. This approximation works for SOTA models and results in scores that are well above chance.
>
>   &nbsp;
>
>   However, models from the Pythia Suite (Biderman, 2023\) are used for the LLM size comparison since the Pythia suite provides a big range of model sizes. Pythia models are not SOTA and show substantially lower downstream competence than more recent (and often distilled) models such as Llama-8B. We therefore expect this gap to impact probing performance, particularly complex tasks like metro lines.
>
>   &nbsp;
>
>    | Model        | Type Score |
>    |--------------|-------------|
>    | Llama-8B     |     0.53  |
>    | Pythia-6.9B  |     0.26  |
>
> ---
>
> * > The paper does not sufficiently motivate that there is a long-standing debate over whether neural networks or connectionist models can understand semantics.
>
>     We agree with the reviewer that motivating earlier the connectionist vs. symbolic debate can amplify the paper’s impact and reach.
>
>   &nbsp;
>
>   We move this paragraph to the introduction and extend it as follows:
>
>     ```
>     Symbolic structures and connectionist models
>
>     Linguistic theory has long held that language is governed by symbolic, combinatorial structures consisting of abstract, tree-like representations that support discrete operations (Fodor & Pylyshyn, 1988; Marcus, 2003). Traditionally, this view has been seen as fundamentally at odds with connectionist models, which operate through continuous, high-dimensional vector representations (Smolensky, 1991; Gayler, 2004). Yet recent advances in large neural networks have complicated this divide: despite lacking explicit symbolic machinery, these models achieve remarkable linguistic performance (Futrell & Mahowald, 2025; Griffiths, 2025). As a result, symbolic and connectionist perspectives now appear closer to reconciliation, highlighting a key open question: how are symbolic representations and operations supported within the vectorial activation spaces of Large Language Models?
>    ```
> &nbsp;
>
> ---
> We hope our response  and additional analyses clarified adequately the reviewer’s concerns and, if so, that this is reflected in the updated score.

---

> > ### Author Response · Authors · 2025-11-22
> >
> > **Citations:**
> > (Hewitt & Liang, 2019\) https://aclanthology.org/D19-1275.pdf
> > (Maudslay, 2021\) https://aclanthology.org/2021.naacl-main.11.pdf
> > (Belinkov, 2022\) https://aclanthology.org/2022.cl-1.7.pdf
> > (Lake & Baroni, 2018) https://arxiv.org/pdf/1711.00350
> > (Hupkes, 2020\) https://arxiv.org/pdf/1908.08351
> > (Lakretz, 2022\) https://aclanthology.org/2022.coling-1.285.pdf
> > (Futrell & Mahowald, 2025\) https://arxiv.org/pdf/2501.17047
> > (Griffiths, 2025\) https://arxiv.org/pdf/2508.05776

---

### Author Response · Authors · 2025-11-27

Dear Reviewers,

Thank you again for your valuable comments.

We shared our responses to the reviews about a week ago. The responses contain further analyses and clarifications of the issues raised.

With about one week left in the discussion period, we wanted to kindly check whether our responses clarified your concerns, and whether there are any remaining points that would benefit from further clarification.

We appreciate your time and feedback, and we welcome any additional remarks.

Best,

---

### Author Response · Authors · 2025-12-02
**Thanks to the reviewers and summary of the rebuttal (1/2)**

We would like to thanks the reviewers and the AC for the efforts put in reviewing our work.

Despite the exceptional circumstances the rebuttal process has been constructive and has helped to refine the initial manuscript.

---

Next, we briefly present (1) the strengths and (2) the weaknesses as well as (3) the additional experiments and clarifications carried out to address the reviewers’ concerns.

&nbsp;


## Strengths

* **Clear hypothesis and precise experiments**

   - > **e9gD**: The key question, “How do artificial neural networks bind concepts to form complex semantic structures?”, is addressed through simple but precise experiments.
   - > **rsDG**: Overall, to the best of my knowledge, the paper is careful in its claims and matches them to what the experiments can actually support and are novel in terms of experiments construction for the hypothesis.
   - > **NUp5**: The authors are careful in their application of the polar probe technique
   - > **FrSx**: The central hypothesis is clear, and the authors test it with a reasonable set of ablations (model size, layer, graph complexity, OOD generalization).

* **Useful application of polar probes to semantic structures**
   - > **rsDG**: I like how precise the core hypothesis in the paper is: existence = distance, type = direction. And I think keeping the probe strictly linear helps me trust the interpretation.
   - > **e9gD**: The experiments show that polar probes are a useful tool for understanding semantics in LLMs, especially for Euclidean graph structures.

* **Convincing and extensive experiments: OOD, emergence thorughout pre-training and correlation with behaviour**
   - >  **rsDG**: The evaluation protocol/setups also look correct to my eyes: the same geometric test across multiple semantic domains strengthens the result. I also like the emergence analyses (layer position, scaling etc.). The OOD checks and the correlation with QA, while not causal, are sensible sanity checks that the decoded geometry aligns with behavior.
   - > **NUp5**: study a wide variety of semantic relationships across a wide variety of model families and sizes and even perform analysis across model checkpoints.
   -> **FrSx**: Link to Downstream Function: The inclusion of a correlation analysis (Figure 6) between probe error and QA performance is a necessary and welcome step, providing preliminary evidence that these representations are not merely epiphenomenal.

* **Well written and clear**
   - > **e9gD**: The paper is well written. Although organized slightly unconventionally, it is easy to follow and read.

---

> ### Author Response · Authors · 2025-12-02
> **Thanks to the reviewers and summary of the rebuttal (2/2)**
>
> ## Weaknesses
>
> ---
>
> * **Experiments are limited to controlled datasets**
>    - > **e9gD**: The toy datasets proposed in this work are a reasonable contribution but might not be sufficient on their own.
>    - > **sDGr**: One potential drawback is the reliance on synthetic prompts. This leaves open whether the same distance–direction code appears in natural text, where relations are implicit and surface forms vary. It would strenghthen author's arguments much more if they can construct or source some evaluation protocol based on natural/free texts.
>
>    &nbsp;
>
>    ### **New experiments**
>
>    We have created two naturalistic datasets (English and Multilingual) to evaluate polar probes beyond controlled datasets. The results show that polar probes trained on a controlled sentences (English only) generalise to naturalistic datasets even in the multilingual case.
>
>    (Figure 14) https://drive.google.com/file/d/14Ft-qaBAHjZjAhXnkHq-odFR-8fPJn7H/view?usp=sharing
>
>    &nbsp;
>
> ---
> * **Lack of causal interventions**
>    - > **FrSx**: The paper is a classic “probing” study. It shows that a linear decoder can be trained to extract information, but it provides no causal evidence that the model actually uses this geometric structure to perform computations.
>
>    &nbsp;
>
>    ### **New experiments**
>
>    We have added a causal intervention analysis where we successfully steer Llama-8B using polar probe prototypes. The result shows that steering works in middle layers, where the performance of polar probes is the highest.
>
>    (Figures 8 and 9; Table 2) https://drive.google.com/file/d/14Ft-qaBAHjZjAhXnkHq-odFR-8fPJn7H/view?usp=sharing
>
>    &nbsp;
>
> ---
> * **Polar probe performance degrades with complexity**
>    - >  **FrSx**: The experiments are confined to “minimalist” semantic structures (e.g., graphs with 4–6 entities). The paper’s own results (Figure 4, middle) show severe performance degradation as the number of entities increases.
>
>    &nbsp;
>
>    ### **Clarification**
>
>    LLMs are known to face challenges on compositional and reasoning tasks (Benchekroun, 2023 Figure 6 ; Dziri, 2023 ; Shojaee, 2025). In our experiments, computational constraints limit us to using 8B models without chain-of-thought reasoning. As a result, probe performance is inherently capped by the LLM’s downstream competence. Extending this analysis to larger reasoning models, and examining how such geometric structures develop throughout the reasoning trace, constitutes a promising avenue for future work.
>
> ---
>
>
>    &nbsp;
>
> * **Previous work on spatial world models**
>    - >  **FrSx**: The core methodology is a direct application of an existing tool (the Polar Probe) to a new set of synthetic datasets. The paper’s central finding — that (spatial) relations are encoded geometrically by direction — is not new. For example, similar work (Tehenan et al., "Linear Spatial World Models Emerge in Large Language Models"), has already demonstrated this finding
>
>    &nbsp;
>
>    ### **Clarification**
>
>    (Tehenan, 2025) differs from our contribution in the following aspects:
>
>      - **Broader scope**: Here we go far beyond modeling 2D or 3D spatial coordinates: and show the viability (and limits) of polar probes on conceptual (and non-euclidean) semantic domains such as Thematic roles, Family trees and Metro maps.
>      - **Methodological difference**: (Tehenan, 2025) regresses model activations onto ground-truth 2D or 3D coordinates similarily to (Gurnee, 2024). This approach cannot be generalized to conceptual semantic graphs, where there exists no absolute ground-truth coordinates. Conceptual graphs contain only relative relational structure, which in turn, can be captured by polar probes.
>    &nbsp;
>
>
> ---
>
> Once again, we would like to thank the reviewers and ACs for their time.
>
> We hope this summary clarifies the strengths and weaknesses pointed by the reviewers and how we have addressed the reviewers' comments. Further clarifications and experiments are described in more detail in the full rebuttal below.

---

### Meta-Review · Area_Chair_Pc5y · 2026-01-04

**Summary:**

### Minimalist datasets
* Multiple reviewers noted the simplicity of the datasets examined (e.g. "One potential drawback is the reliance on synthetic prompts"). The authors attempted to respond to this by showing generalization to a "naturalistic" dataset of LLM-generated text. However, there are no details about how this text was created, or examples of the kinds of text that came out of the LLMs. As a result, it is hard to interpret this experiment, and it does not address the concerns of the reviewers.

### Novelty with respect to existing work
* Multiple reviewers noted similarity to previous work, both in the probe technique already existing, and in the results already being partially understood (for syntactic tasks, and for semantic tasks which involve spatial relations in 2D or 3D). With the similarity of these components to prior work being so high, there needs to be more justification for why semantic relationships as a task is significantly different from what has already been analyzed, or how the results here can be integrated with those that already exist in these other closely related domains.

### Scaling issues
* Reviewer FrSx noted that the experiments only examined graphs with up to 6 entities, with performance of the probes also dropping when the number of entities was increased. The authors responded claiming that the base LLMs examined can only reason over this many entities, which restricts the scope. However, it is not clear that this is the case. Evidence should be provided that the polar probing technique works __as long as the base LLM also works__. If LLM performance does not degrade with more than 6 entities , but polar probes do (or the other way around), that is important to understand.

As a result of these concerns, I recommend rejecting this paper, but I believe that fully integrating the new results and contextualizing better with related work will make for a strong future submission.

**Reviewer Concerns:**

### Addressed
* Causal intervention experiments (as requested) show that the geometric structures uncovered by polar probes are not just useful for correlation analysis, but can also be used for causal steering as well, and that this works best at middle layers.
* Further experimental analyses, including a study of uncontextualized embeddings.

### Unaddressed
* Scaling to graphs of larger sizes.
* Generalization to more naturalistic datasets.
* Contextualizing in related work beyond pointing out distinctions (how do the results here either mesh with or go against related studies?)

**Reviewer Scores:**

Reviewer e9gD
* would not have changed their score. They voted to accept, and one of their key requests (an additional study on non-Euclidean geometry) was not performed.

Reviewer sDGr
* would not have changed their score. A main concern was on the synthetic dataset extension, and there were no details provided for the naturalistic dataset which the authors studied in the rebuttal.

Reviewer NUp5
* would not have changed their score. Central concern was impact of the results given their predictability from prior work, which was unaddressed in the rebuttal.

Reviewer FrSx
* may have raised their score to a 4, but unlikely to have changed to an accept, given that many of their questions were unaddressed by the authors.

---

### Decision · Program_Chairs · 2026-01-26

Reject